# Lead Mobilization and Speciation in Mining Waste: Experiments and Modeling

**Clémentine Drapeau [1], Rabei Argane [2,3], Cécile Delolme [4], Denise Blanc [5], Mostafa Benzaazoua [2,6], Rachid Hakkou [3,6], Thomas Baumgartl [7], Mansour Edraki [8] and Laurent Lassabatere [9,*]**

1   Direction Régionale de L'environnement, de L'aménagement et du Logement (DREAL), Auvergne Rhône-Alpes, Unité Départementale du Rhône, 63 Avenue Roger Salengro, 69100 Villeurbanne, France; clementine.drapeau@developpement-durable.gouv.fr
2   Research Institute of Mines and Environment, Université du Québec en Abitibi-Témiscamingue, 445 boul. de l'Université, Rouyn-Noranda, QC J9X 5E4, Canada; rabei.argane@uqat.ca (R.A.); Mostafa.Benzaazoua@uqat.ca (M.B.)
3   Imed-Lab, Faculty of Sciences and Technology, Cadi Ayyad University (UCA), Abdelkarim Elkhattabi Avenue, Gueliz, P.O. Box 549, Marrakech 40000, Morocco; r.hakkou@uca.ma
4   Université de Lyon, ENTPE, 3 Rue Maurice Audin, 69518 Vaulx-en-Velin, France; cecile.delolme@entpe.fr
5   Déchets Eau Environnement Pollutions (DEEP), INSA-Lyon, Université de Lyon, 7 Rue de la Physique, 69621 Villeurbanne, France; denise.blanc@insa-lyon.fr
6   Mining Environment & Cicrular Economy Program (EMEC), Mohammed VI Polytechnic University (UM6P), Lot 660.Hay Moulay Rachid, Ben Guerir 43150, Morocco
7   Geotechnical and Hydrogeological Engineering Research Group (GHERG), Federation University Australia, Churchill, VIC 3841, Australia; t.baumgartl@federation.edu.au
8   Centre for Mined Land Rehabilitation, Sustainable Minerals Institute (SMI), The University of Queensland, St. Lucia, QLD 4072, Australia; m.edraki@cmlr.uq.edu.au
9   Université de Lyon, UMR5023 Ecologie des Hydrosystèmes Naturels et Anthropisés, Université Lyon 1, ENTPE, CNRS, 3 Rue Maurice Audin, 69518 Vaulx-en-Velin, France
*   Correspondence: laurent.lassabatere@entpe.fr

**Abstract:** Mining produces significant amounts of solid mineral waste. Mine waste storage facilities are often challenging to manage and may cause environmental problems. Mining waste is often linked to contaminated mine drainage, including acidic waters with more or less elevated concentrations of trace metals such as lead. This work presents a study on the mobilization of lead from waste from two typical mining sites: Zeida and Mibladen, two now-closed former Pb–Zn mines in the Moulouya region of Morocco. Our research investigates the mobilization potential of Pb from the waste of these mines. The study involved acid–base neutralization capacity tests (ANC–BNC) combined with geochemical modeling. Experimental data allowed for the quantification of the buffering capacity of the samples and the mobilization rates of lead as a function of pH. The geochemical model was fitted to experimental results with thermodynamic considerations. The geochemical model allowed for the identification of the mineral phases involved in providing the buffering capacity of carbonated mining waste (Mibladen) and the meager buffering capacity of the silicate mining waste (Zeida). These cases are representative of contaminated neutral drainage (CND) and acid mine drainage (AMD), respectively. The results highlight the consistency between the ANC–BNC experimental data and the associated modeling in terms of geochemical behavior, validating the approach and identifying the main mechanisms involved. The modeling approach identifies the dissolution of the main solid phases, which impact the pH and the speciation of lead as a function of the pH. This innovative approach, combining ANC–BNC experiments and geochemical modeling, allowed for the accurate identification of mineral phases and surface complexation phenomena, which control the release of lead and its speciation in drainage solutions, as well as within solid phases, as a function of pH.

**Keywords:** mining drainage; lead; mobilization; modeling; speciation

## 1. Introduction

Open-pit and underground mining operations produce a significant amount of solid mineral waste. Whether it is mine waste rock in the form of waste rock piles or concentrator rejects discharged in the form of tailings, above-ground waste storage facilities are difficult to manage and expensive to rehabilitate. These voluminous and unique discharges are the origin of many environmental problems worldwide because they generate effluents containing significant quantities of trace metals, sometimes accompanied by strong acidification of the environment [1,2]. Acid mine drainage (AMD) represents the phenomenon of acidification of the environment due to the oxidation of sulfidic mining waste, while contaminated neutral drainage (CND) refers to drainage from mining waste with acid-neutralizing capacity and water pH close to neutrality. Both water types may have high concentrations of metals, metalloids, and sulfate [3,4].

In Morocco, the mining industry has always been an important contributor to the economy. This sector represents more than 27% of all exports and nearly 6% of gross domestic product [1]. Unfortunately, this industry has too often ignored the need to rehabilitate mining sites in the past, including tailing storage facilities and waste rock dumps. More than 200 abandoned sites have been identified across the country [5,6]. The intense mining activity of the Zeida (1972–1985) and Mibladen (1938–1983) mines in the upper Moulouya region generated large volumes of mining waste in the form of waste rock and mining residues from ore concentration stages. They are also used as uncontrolled construction materials: finishing and surfacing mortar for residential walls [1,7,8]. After the cessation of operations, this mining waste significantly contaminated water resources and adjacent soils [9–12]. Indeed, in addition to the substantial contamination of surface waters and soils, the weathering of this waste triggers the release of some pollutants, including toxic trace metals, e.g., [13–15].

In this context, understanding weathering processes and related contaminated neutral drainage (CND) or acid mine drainage (AMD) is fundamental for quantifying trace metal release into the environment and related environmental issues. The mobilization of trace metals is complex. It depends on geochemical processes at the solid/liquid interface: precipitation/dissolution, ion exchange, surface complexation, surface precipitation, sorption processes driven by water repellency, absorption (incorporation of the solute in the solid matrix), and diffusion [16]. All of these interactions are featured by the chemical, physicochemical, mineralogical, and physical properties of water, dissolved species, and surfaces. pH remains one of the key factors controlling the dissolution and desorption of mineral elements [16–18].

Numerous studies have provided information on the total concentrations of trace metals, and the role of physicochemical conditions on pollutant release via batch approaches or selective extractions [18–25]. The characterization of trace metal speciation is still carried out sporadically by micro-analysis or indirectly by sequential extractions. However, the environmental and human health risks depend not only on the total content but also on the trace metal speciation [26–28], and there is still a lack of information on the main solid phases and geochemical processes that control the potential mobility of major elements and trace metals in these complex matrices.

This paper presents a study on lead release from mining waste due to pH change for the mines of Zeida and Mibladen located in Morocco. We aimed at identifying the geochemical mechanisms responsible for the lead release and speciation in surface waters from those sites. Geochemical models are built as the first step towards modeling the fate of trace metals for operational objectives, including (i) better management of large quantities of mining waste, (ii) providing information on the risk of chemical contamination of the waste repository in the surrounding environment, and (iii) improving analysis of the reuse of waste for the construction sector. This study combines experimental data and geochemical modeling to fill in the gap of understanding of lead release in mining waste [17,18,29]. The experimental part is mainly based on geochemical characterization and acid–base neutralization capacity tests (ANC–BNC). The modeling approach, making

use of PHREEQC geochemical software, is calibrated against experimental data from the ANC–BNC tests to simulate lead release and speciation in the liquid phase as a function of pH [30]. Lead was chosen, since it enlists heavy metals with high potential risks to human health [31], and it has been found at significant concentrations in the two studied mine tailings [1,32] with potential risks for the surrounding and downstream environments [33]. These steps allowed for the identification of the main phases bearing lead in the two studied types of mining waste and the main mechanisms responsible for their dissolution and related lead release.

## 2. Materials and Methods

### 2.1. Sampling Site, Geochemical Characterization, and ANC–BNC Tests

Mining waste characterization has already been performed by Argane et al. [1], and more details can be found in their paper. The studied samples were taken from the "tailings pond" of the Zeida and Mibladen mines in the Moulouya region of Morocco, used by locals as a supply of building materials in a completely uncontrolled manner. Large amounts of samples were extracted for this study from below the surface to avoid weathered material. Several representative locations of the tailing ponds were sampled. The climate of the region is semiarid, the annual precipitation varies from 100 to 400 mm, and the mean annual temperature ranges from 12 to 14 °C [34]. The samples (five from Zeida and two from Mibladen) correspond to composite samples over 1 m and were taken with a mechanical shovel over an area of approximately 1 m × 2 m before being dried and homogenized and before geochemical characterization [7,32].

The samples were analyzed by X-ray diffraction (XRD) in order to determine the proportions of the crystallized mineral phases. XRD analyses were performed using a Bruker A.X.S D8 Advance diffractometer equipped with a copper anticathode that scans over a diffraction angle (2θ) from 5° to 60°. Scan settings were a 0.005° 2θ step size and a 1 s counting time per step. The "Diffrac-Plus" EVA software was used to identify mineral species, and "TOPAS" software implementing Rietveld refinement allowed for the quantification of the abundance of all identified mineral species. The samples (polished sections) were also observed with a scanning electron microscope (SEM) using backscattered electrons (BSEs). The microscope was a Hitachi S-3500N variable pressure microscope equipped with an X-ray energy dispersive spectrometer from Oxford (EDS; Silicon drift spectrometer X-Max 20 mm$^2$). The operating conditions involved an electron beam of 20 keV, z100 mA, and a 15 mm working distance. The INCA software provided spectra and X mapping. More details can be found in [7,32].

### 2.2. ANC–BNC Experiments

The acid–base neutralization capacity test (ANC–BNC) was designed to quantify the mobilization of pollutants and buffering capacities of solid matrices in contact with a source of protons and alkalinity [18,35–37]. In this study, we used the experimental results from Argane et al. [1]. These authors performed batch tests according to CEN/TS 14,429 [38]. The test involved a series of extractions using acid ($HNO_3$) and base (NaOH) solutions of varying concentrations to cover the pH range between 2 and 12, as suggested by Chatain et al. [17]. In each batch test, solid samples (10 g) were put into contact with 100 mL solutions (liquid/solid ratio of 10 mL/g), and the reactors were agitated for 8 days using an end-over-end tumbler. Such contact time allowed for the attainment of stabilized pH values. Leachates were then filtered with 0.45 μm membranes before pH measurement. The samples then underwent acidification before ICP-AES analyses to determine trace metal concentrations. More details can be found in [7,32].

The ANC–BNC results were treated as follows. We determined the amount of protons ($H^+$) added for the acid part (ANC) or the amount of OH− multiplied by −1 for the basic part (BNC). Eq_$H^+$ has an algebraic value determining the inputs of acidity or alkalinity to the systems. The experimental curve plotted the evolution of pH as a function of the added $H^+$, Eq_$H^+$ [17]. The pH (Eq_$H^+$) curve was then used to compute a transformed

curve by differentiating the inverse function Eq_H$^+$ (pH) with regard to pH [36,37]. The transformed curve helps to determine the buffer zones and to relate these to the dissolution of specific mineral phases, thus allowing for a better understanding of element release and mineral dissolution [37]. Such an experimental approach was also validated for the characterization of AMD and CND using synthetic pure mineral assemblages [39].

*2.3. Geochemical Modelling*

The geochemical modeling was performed using PHREEQC software (Interactive, version alpha 3.1.2.8538, developed by USGS, United States Geological Survey), making it possible to simulate the ANC–BNC experiments conceptually [37,39–41]. PHREEQC is based on a deterministic approach where digital tools solve all chemical reactions and associated equilibrium equations [30]. In this study, the *llnl* (Lawrence Livermore National Laboratory) database was considered. It is one of the international reference databases that is very well documented and contains all the studied mineral phases. The geochemical modeling presented in this article has been adjusted to the ANC–BNC experimental results to select the geochemical processes involved in the release of the elements and predict the element speciation as a function of pH.

ANC–BNC experiments were modeled as closed batch reactors, considering the actual conditions of experimentation, i.e., the actual mass of solid, the actual volume of solution, and the actual air volume in the closed reactors. The latter was deduced from the knowledge of the volumes of the solutions and the vessels. The system was then characterized by its gas/liquid ratio (the "Vol_Air_fix" parameter required by PHREEQC). The gas phase's initial pressure was set at atmospheric pressure at sea level, corresponding to a ratio pressure/reference pressure of unity (the "value of P_Air_0" parameter required by PHREEQC). The initial partial pressures of nitrogen, oxygen, and carbon dioxide were fixed at their typical atmospheric values. The mineral assemblages were simulated as the combination of solids with related initial amounts and solubility constants, as defined by the *llnl* database. The mixture of solid, liquid, and gas phases was then simulated for the solutions used to perform ANC–BNC experiments: neutral (water at pH 7), acidic (an addition of HNO$_3$), or alkaline (an addition of NaOH). PHREEQC numerically resolved the system considering mass conservation, the law of mass action, and Henry's law for mass transfer between the liquid and gas phases.

Regarding optimization, we multiplied modeling scenarios corresponding to variable mineral compositions (in terms of selected phases and initial amounts) and the variable compositions of sorption sites in terms of site density and sorption constants, with ~700 simulations for the Mibladen site and ~900 for the Zeida site. These numerous iterations allowed us to conduct sensitivity analysis, to identify critical parameters, to identify compensation effects, and to invert the observations (final fit). These also allowed for the definition of an optimum model after the following steps: (i) an initial estimate of the composition of the mineral assemblages (solid samples) and the gas phase with an indication of the minerals' propensity to dissolve or precipitate; (ii) modeling of this assemblage in contact with the acid and alkaline solutions, assuming chemical equilibrium; (iii) a comparison of predicted against observed pH and element concentrations; (iv) a trial-and-error-based forward–backward procedure, with a repetition of steps (i)–(iii) until a consistent model with accurate fits could be obtained; and finally, (v) the identification of the more likely mineral composition and a report of modeled element speciation in the liquid and solid phases as a function of pH. Note that the same initial mineral composition is considered for all acid and alkaline solutions (i.e., for all ANC–BNC experiments).

Regarding the quality of the fit, two goodness of fit indicators were considered. First, the coefficient of determination, $R^2$, was computed as follows:

$$R^2 = \left\{ \frac{\sum_{t=1}^{N} (y_{o,t} - \overline{y_o})(y_{s,t} - \overline{y_s})}{\left[ \sum_{t=1}^{N} (y_{o,t} - \overline{y_o})^2 \right]^{0.5} \left[ \sum_{t=1}^{N} (y_{s,t} - \overline{y_s})^2 \right]^{0.5}} \right\}^2 \tag{1}$$

where $N$ is the number of observations; $y_{s,t}$ and $y_{0,t}$ are the simulated (modeled) and observed values at instant $t$, respectively; and, $\overline{y_s}$ and $\overline{y_o}$ are their respective means. The normalized root mean square error (NRMSE) was also computed, as follows [42]:

$$NRMSE = abs \left\{ \frac{1}{\overline{y_o}} \left[ \frac{1}{N} \sum_{t=1}^{N} (y_{s,t} - y_{o,t})^2 \right]^{\frac{1}{2}} \right\} \tag{2}$$

The coefficient of determination quantifies the correlation between the modeled and observed data. The NRMSE quantifies the errors between modeled and observed data.

## 3. Results and Discussion

### 3.1. Preliminary Results

#### 3.1.1. Characterization of the Mining Waste

The results of the XRD diffraction (Table 1) reveal that the two mine tailings have high concentrations of silicates: (in decreasing order) quartz, orthoclase, albite, chamosite, and kaolinite at Zeida and only quartz and chamosite at Mibladen. Barite and fluorite could also be found in the two mine tailings. Note that the XRD analysis does not detect the amorphous phases. However, we assumed that these results gave good insight into the relative proportions of all types of minerals, including both crystallized and amorphous minerals. The main difference between the two sites relates to their carbonate content, with significant concentrations at the Mibladen site versus zero content at the Zeida site (see Table 1).

**Table 1.** Mineral characterization and mass percentage of the different crystallized phases as determined by X-ray diffraction (in % of dry matter, *w/w*) (adapted from [32], amorphous phases undetected).

| Mineral Phase | | Formula | % Zeida | % Mibladen |
|---|---|---|---|---|
| Silicates | Quartz | $SiO_2$ | 48.8 | 7.16 |
| | Orthoclase | $KAlSi_3O_8$ | 31.6 | - |
| | Chamosite | $Fe_2Al_2SiO_5(OH)_4$ | 2.6 | 2.72 |
| | Albite | $NaAlSi_3O_8$ | 5.5 | - |
| | Kaolinite | $Al_2Si_2O_5(OH)_4$ | 1.5 | - |
| Carbonates | Dolomite | $CaMg(CO_3)_2$ | - | 51.35 |
| | Calcite | $CaCO_3$ | - | 22.2 |
| Others | Barite | $BaSO_4$ | 7.9 | 16.57 |
| | Fluorite | $CaF_2$ | 2.1 | 3 |

SEM analysis provided additional information on the phases. First, SEM analyses confirmed the presence of the minerals found with XRD, with, for instance, the occurrence of grains of quartz, barite, and fluorite, illustrated in Figure 1a. SEM analyses also revealed the presence of minerals that were not detected using XRD. For instance, small grains of pyrite could be found, often in association with silicate phases (Figure 1a). We expect that pyrite concentrations were too low to be detectable using XRD analyses (concentrations probably lower than 1%). From the presence of pyrite in the two mine tailings, as revealed by SEM analyses, we can conclude that the waste from Zeida is representative of mining waste prone to the acid mine drainage (AMD) process. Conversely, Mibladen mining waste is expected to have a substantial buffering capacity and therefore neutralize the acidity produced by pyrite oxidation and dissolution, leading to a contaminated neutral drainage (CND) process.

Regarding trace elements and trace metals, SEM also provided relevant insights by revealing phases that were not detected with XRD analyses. For lead, SEM analyses detected mineral forms of massicot (Figure 1b), galena (Figure 1c), cerussite in Zeida mine tailings and of galena, and cerussite in Mibladen mine tailings (Figure 1d). More details and

images can be found in [7]. The concentrations of the crystalline forms of these minerals may be too low to be detected with X-ray diffraction (<1%).

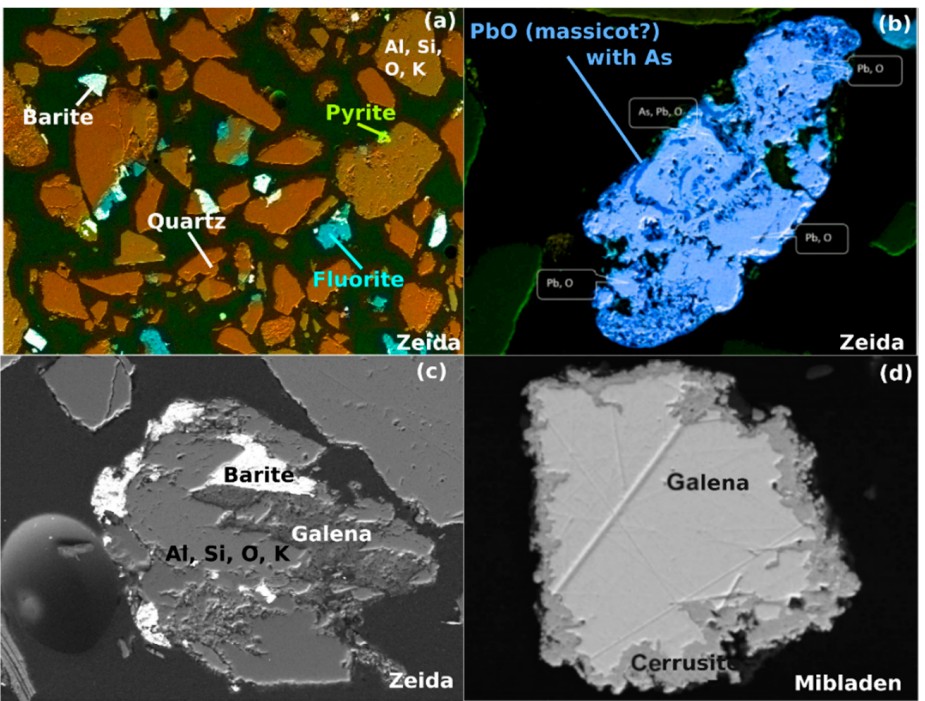

**Figure 1.** SEM images of the mining tailings at the Zeida and Mibladen sites; minerals are indicated if correctly identified, and the atomic composition is provided otherwise, (**a**) global view of Zeida mining waste, (**b**) grain of massicot, (**c**) grains of galena and barite in association with aluminosilicates, (**d**) grain of galena which alteration provide cerussite.

### 3.1.2. Modeling Approach and Validation of the Geochemical Model

The geochemical model was based on the results detailed in the previous section and, particularly, on the lists of detected minerals using XRD analyses and SEM observations. In addition to that list, the analysis of the different buffer capacities as a function of pH was analyzed to derive the main constituents of the mineral assemblages on the basis of previous works [37,39]. This method is illustrated below. In addition, the minerals expected to form, precipitate, or dissolve given thermodynamic considerations and preliminary modeling results were also added. The final selection of the mineral composition involved in the geochemical processes requires a forward–backward approach, combining modeling, mineralogical characterization, and literature. The main equations impacting element speciation in the solid, liquid, and gas phases are described in Table 2.

For lead, we considered galena ($PbS_{(s)}$) and cerussite ($PbCO_{3(s)}$), as the SEM analysis revealed them. Alamosite ($PbSiO_{3(s)}$) and lanarkite ($Pb_2(SO_4)O_{(s)}$) had to be added to allow accurate fits of lead release, even if these were not identified using XRD or SEM (Table 2, IM). The fit of lead release allowed us to characterize the lead sorption in terms of cationic exchange and surface complexation onto iron hydroxides ($Fe(OH)_{3(s)}$). In addition to sorption/desorption processes, lead may be released when the phases bearing the sorption sites dissolve (such as iron hydroxides). Note that only the site density and the amounts of minerals were optimized, whereas constants for the Pb complexation were unchanged from the *llnl* database (considering the non-electrostatic model).

**Table 2.** Mineral assemblages obtained by inverse modeling (the "Phases" column), dissolution/precipitation reactions with thermodynamic constants ("Chemical Reactions" and "Equilibrium Constants"), and initial concentrations in the assemblage; surface complexation and cationic exchange sites are described in the three bottom rows. For thermodynamic constants, refer to the *llnl* database [30].

| Phases | Chemical Reactions | Equilibrium Constants (pK) | Initial Concentrations (mol/L) | |
|---|---|---|---|---|
| | | | Zeida | Mibladen |
| *Main Mineral Phases* | | | | |
| *Carbonate* | | | | |
| Calcite | $CaCO_{3(s)} + H^+ \leftrightarrow Ca^{2+} + HCO_3^-$ | 1.8487 | - | 0.014326 (MC) |
| | Content of Pb in inclusion | | - | 0.001188 (SEM/EDS) |
| Dolomite | $CaMg(CO_3)_{2(s)} + 2 H^+ \leftrightarrow Ca^{2+} + Mg^{2+} + 2 HCO_3^-$ | 2.5135 | - | 0.0222198 (MC) |
| *Silicate* | | | | |
| Albite (DO) | $NaAlSi_3O_{8(s)} + 4H^+ \leftrightarrow Al^{3+} + Na^+ + 2H_2O + 3SiO_2$ | 2.7645 | 0.0003013 (MC) | - |
| Chamosite (DO) | $Fe_2Al_2SiO_5(OH)_{4(s)} + 10H^+ \leftrightarrow SiO_2 + 2Al^{3+} + 2Fe^{2+} + 7 H_2O$ | 32.8416 | 0.0002005 (MC) | 0.0003767 (MC) |
| Kaolinite (DO) | $Al_2Si_2O_5(OH)_{4(s)} + 6H^+ \leftrightarrow 2 Al^{3+} + 2 SiO_2 + 5 H_2O$ | 6.8101 | 0.001822 (MC) | - |
| Quartz (DO) | $SiO_{2(s)} \leftrightarrow SiO_2$ | −3.9993 | 0.0129952 (MC) | 0.008307 (MC) |
| Orthoclase (DO) | $KAlSi_3O_{8(s)} + 4H^+ \leftrightarrow Al^{3+} + K^+ + 2H_2O + 3SiO_2$ | −0.2753 | 0.00908 (MC) | - |
| *Others* | | | | |
| Pyrite | $FeS_{2(s)} + H_2O \leftrightarrow 0.25H^+ + 0.25SO_4^{2-} + Fe^{2+} + 1.75HS^-$ | −24.6534 | 0.0001 (SEM/EDS) | 0.000025 (SEM/EDS) |
| Barite | $BaSO_{4(s)} \leftrightarrow Ba^{2+} + SO_4^{2-}$ | −9.9711 | 0.003385 (MC) | 0.00284 (MC) |
| Fluorite | $CaF_{2(s)} \leftrightarrow Ca^{2+} + 2 F^-$ | −10.0370 | 0.000269 (MC) | 0.00457 (MC) |
| Iron hydroxydes | $Fe(OH)_{3(s)} + 3 H^+ \leftrightarrow Fe^{3+} + 3 H_2O$ | 5.6556 | 0.001 (SEM/EDS) | 0.00743 (SEM/EDS) |
| Hematite (DO) | $Fe_2O_{3(s)} + 6 H^+ \leftrightarrow 2 Fe^{3+} + 3 H_2O$ | 0.1086 | 0.0001 (IM) | 0.001 (IM) |
| Fluorapatite | $Ca_5(PO_4)_3F_{(s)} + 3 H^+ \leftrightarrow F^- + 3 HPO_4^{2-} + 5 Ca^{2+}$ | −24.9940 | 0.0001 (IM) | - |
| *Gas phases* | | | | |
| $CO_2(g)$ | $CO_{2(g)} + H_2O \leftrightarrow H^+ + HCO_3^-$ | Vol_Air_fix * = 2.5 | In equilibrium with neutral batch solution | |
| $O_2(g)$ | $NO_{2(g)} + 0.5H_2O + 0.25O_{2(g)} \leftrightarrow H^+ + NO_3^-$ | P_Air_0 ** = 1 | | |
| *Mineral phases bearing lead* | | | | |
| Alamosite (DO) | $PbSiO_{3(s)} + 2 H^+ \leftrightarrow H_2O + Pb^{2+} + SiO_2$ | 5.6733 | 0.00043 (IM) | 0.00000396 (IM) |
| Cerussite (DO) | $PbCO_{3(s)} + H^+ \leftrightarrow HCO_3^- + Pb^{2+}$ | −3.2091 | 0.00009 (SEM/EDS) | $1.098 \times 10^{-5}$ (SEM/EDS) |
| Galena (DO) | $PbS_{(s)} + H^+ \leftrightarrow HS^- + Pb^{2+}$ | −14.8544 | 0.00071 (SEM/EDS) | $9.794 \times 10^{-6}$ (SEM/EDS) |
| Pb (DO) | $P + 2 H^+ + 0.5 O_2 \leftrightarrow H_2O + Pb^{2+}$ | 47.1871 | - | $2.9 \times 10^{-6}$ (SEM/EDS) |
| Lanarkite | $Pb_2(SO_4)O_{(s)} + 2 H^+ \leftrightarrow H_2O + SO_4^{2+} + 2 Pb^{2+}$ | −0.4692 | 0.000001 (IM) | - |
| *Surface complexation site for sorbing lead* | | | | |
| $\equiv Hfo\_sOH$ | $\equiv Hfo\_sOH + Pb^{2+} \leftrightarrow \equiv Hfo\_sOPb^+ + H^+$ | 4.65 | 0.5 | 0.002 |
| $\equiv Hfo\_wOH$ | $\equiv Hfo\_wOH + Pb^{2+} \leftrightarrow \equiv Hfo\_wOPb^+ + H^+$ | 0.3 | 1.3 | 0.0015 |
| *X (ion exchange) sorption site for lead* | | | | |
| $X^-$ | $Pb^{2+} + 2 X^- \leftrightarrow PbX_2$ | 1.05 | 0.009 | 0.0003 |

(DO): dissolutions only; (MC): mineral characterization; (IM): inverse modeling. * Fixed air volume in contact with 1 L of water. ** Air partial pressure initial value (set at the atmospheric pressure).

According to Biondi et al. [42], various points need to be considered to assess the models' relevance and performance objectively. First, the fits must be of good quality, guaranteeing the goodness of fit. Second, the final inverse solution must rely on a unique set of selected equations and related parameters. Indeed, the uniqueness and plausibility of the estimates guarantee the consistency and soundness of the proposed model, e.g., [43]. In our case, these two main criteria are fulfilled. The coefficients of determination $R^2$ are close to 1, always between 0.729 and 0.965, and the NRMSE scored values are always below 25% (Table 3). As shown in Sections 3.2 and 3.3, pH and lead release are somewhat less well resolved in the Zeida case. However, the two fits can be considered accurate. Regarding non-uniqueness, no other variants (based on alternative hypotheses regarding mineral composition, selected dissolution, and sorption processes) can improve the proposed simulations. The sensitivity analysis proved that the proposed model was among the most likely models.

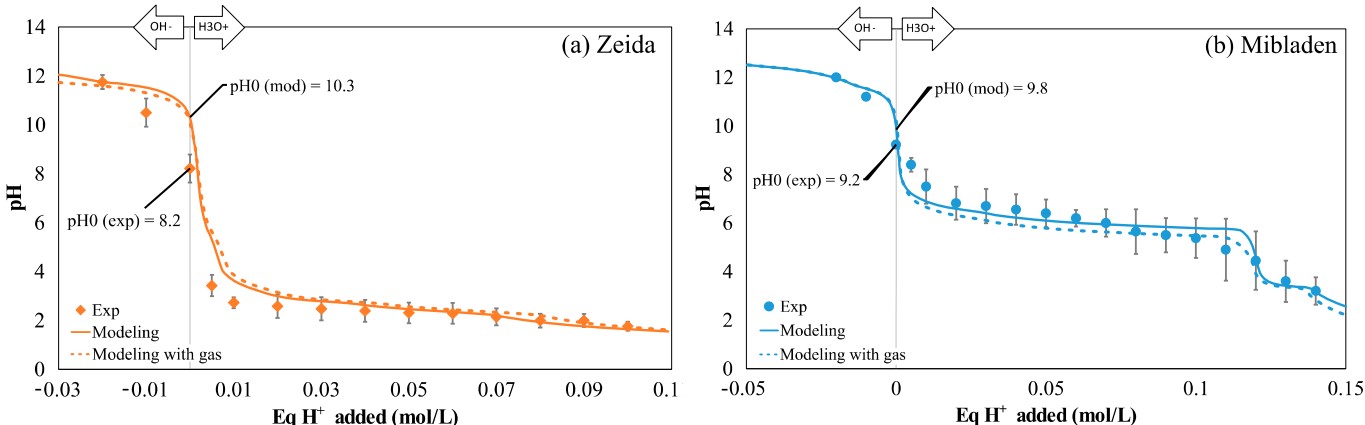

**Figure 2.** Evolution of the pH as a function of the H$^+$ added to the mining waste from Zeida (**a**) and Mibladen (**b**): experimental points (marks, adapted from [1,32])) and geochemical modeling (solid lines indicate the modeling with the gas phase, and the dashed lines indicate the modeling without the gas phase).

**Table 3.** Performance measurement of the final geochemical model.

| Figure Number | Modeled data | NRMSE-Normalization Root Mean Square Error | $R^2$ Coefficient of Determination |
|---|---|---|---|
| Figure 2a | pH = f(Eq H$^+$ added) for Zeida tails | 21.6% | 0.965 |
| Figure 2b | pH = f(Eq H$^+$ added) for Mibladen tails | 7.50% | 0.963 |
| Figure 4a | Pb = f(pH) for Zeida tails | 18.6% | 0.729 |
| Figure 4b | Pb = f(pH) for Mibladen tails | 7.11% | 0.825 |
| Zeida Tails Mean | | 20.1% | 0.847 |
| Mibladen Tails Mean | | 7.30% | 0.894 |

Since all observations were accurately fitted, including pH and element release, and since the proposed model seems unique, we assumed that the modeled results are indicative of the mineral assemblages and the geochemical processes involved in the closed batch reactors. Consequently, the modeled results were used to conclude on processes and lead speciation in the liquid and solid phases.

### 3.2. Characterization of Buffer Capacity with ANC–BNC Experiments

3.2.1. Observed and Modeled Buffer Capacity

Figure 2 shows the change of pH as a function of the added H$^+$ (Eq_H$^+$) corresponding to the contact with all acid and alkaline solutions for the two mine tailings of Zeida (Figure 2a) and Mibladen (Figure 2b). In each of the graphs, the markers denote the experimental results of the ANC–BNC tests, whereas the lines denote the geochemical

models. The experimental value of pH at equilibrium with water has a lower value for Zeida, with a pH of 8.2 versus 9.2 for Mibladen. In the acidic zone, the pH decreases much more in the Zeida case, with values dropping to 2 (Figure 2a), whereas the decrease is much more progressive in the Mibladen case with a plateau around 6 (Figure 2b).

These differences result from the composition of the waste from the two mining sites. The mining waste from the Zeida site is more than 90% (*w*/*w*) silicates, whereas that of the Mibladen mine is less than 10% (*w*/*w*) silicates but almost 75% carbonates (Table 1). The modeling helps us to understand the role of carbonates in the bulk buffer capacity and relates the contrasting evolution of pH to the mineral compositions of the waste from the two mining sites. Indeed, each buffer zone corresponds to the dissolution of a specific mineral. The selection of the proper minerals dissolved at a given pH with appropriate amounts allows for accurate modeling of the curve pH (Eq_H$^+$) (see below, Figure 2) and the identification of the geochemical mechanisms involved. As proposed in previous work [37,39], the transformed pH curve d Eq_H$^+$/d pH as a function of pH improves the identification of minerals. This point is discussed in more detail below (Section 3.2.2).

The geochemical modeling clarified the importance of the gas phase and its components in closed batch reactors. The gas phase's contribution is usually neglected in geochemical approaches, e.g., [37], whereas it may interact with the geochemical processes. Indeed, several of its constituents may significantly impact the element speciation and distribution between the solid, liquid, and gas phases. At first, oxygen may interfere with redox processes and thus with the oxidation of pyrite and induced acidification. In addition, other species in the gas phase may interact with solutes and species in water. The dissolution of $CO_{2(g)}$ produces carbonic acid ($H_2CO_3$) that acidifies the water by transforming into the bicarbonate ion ($HCO_3^-$). This may, in turn, promote the dissolution of carbonated phases, including those involving trace metals, as detailed for lead in the following:

$$CO_{2(g)} + H_2O \Leftrightarrow H^+ + HCO_3^- \tag{3}$$

$$PbCO_{3(s)} + CO_{2(g)} + H_2O \Leftrightarrow 2\,HCO_3^- + Pb^{2+} \tag{4}$$

The geochemical modeling was then performed with and without the gas phase for the specific case of our batch reactors. The comparison of the modeled results with and without the gas phase showed no significant differences. The difference is slightly more visible for the Mibladen site, with lower pH values in the absence of gas. This result shows that neglecting the gas phase may not change the results drastically, at least for reactors in which the gas and the liquid phases have similar volumes. However, if the batch reactors are opened, the geochemical model should consider this contact with an infinite atmospheric reservoir. In the following, the results correspond to the modeling with the gas phase, its main constituents, and the related features (partial pressures and the volume of gas in the reactors).

### 3.2.2. Identification of the Main Mineral Phases by Modeling Buffer Capacity

Previous results have allowed for a global understanding of the impact of mining waste on pH as a function of the added Eq_H$^+$. The model makes it possible to refine these results by determining the effect of each of the significant phases on pH. Indeed, the models include the buffer capacity, the element release, and the solid phase simultaneously, allowing us to establish correspondence between these variables. The modeled results were obtained with the inverse method detailed in Section 3.1.2, and are depicted in Figure 3. Related mineral composition and amounts are described in Table 2. Geochemical modeling allows for the superposition of the transformed curves (derivatives d Eq_H$^+$/d pH) with the solid phase composition (minerals).

Prior to modeling, the buffer curves were derived from Figure 2 by transforming the curve pH (Eq_H$^+$), i.e., by differentiating the experimental curve Eq_H$^+$ (pH). The peaks more precisely delineate the different buffer zones (Figure 3). For the Zeida and Mibladen cases, the buffer zones are mainly in the acidic zone (Z1 and Z2 in Figure 3a

and Z4, Z5, and Z6 in Figure 3b). These values of pH are specific to the matrices and their mineral compositions. In the alkaline area, a similar slight buffering capacity is present at pH $\simeq$ 11.5 in waste from both sites (Z3 in Figure 3a and Z7 in Figure 3b).

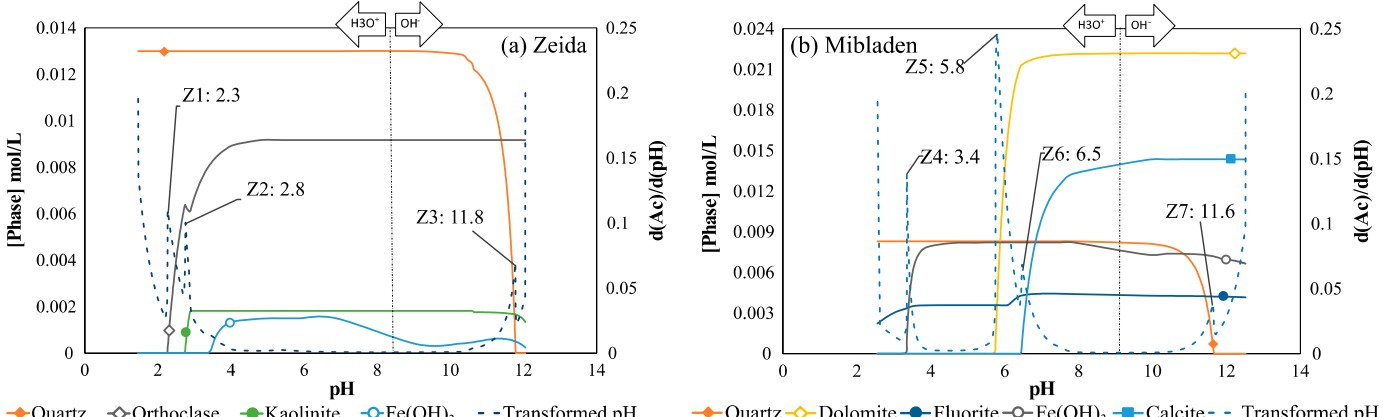

**Figure 3.** Transformed curve (d Eq_H$^+$/d pH = d (Ac)/d (pH), right axis) as a function of pH obtained from observations, along with modeled compositions of mineral phases (left axis), in the Zeida (**a**) and Mibladen (**b**) mining waste.

The position of the buffer zones can be related to a specific mineral, and it is possible to identify the effect of each of the mineral phases on the pH by analyzing the modeled results (see Figure 3a,b, "Quartz" versus "Transformed curve"). Thus, in both the Zeida and the Mibladen mining waste, the buffering capacity in the basic zones (Z3 and Z7) is linked to the dissolution of part of the quartz. The modeled data proves that these zones correspond to the following equation:

$$SiO_{2(s)} + H_2O \Leftrightarrow HSiO_3{}^- + H^+ \tag{5}$$

The main phases corresponding to the observed buffer zones in the acidic zone of the Zeida mining waste are orthoclase (KAlSi$_3$O$_{8(s)}$), kaolinite (Al$_2$Si$_2$O$_5$(OH)$_{4(s)}$), and iron hydroxides (Fe(OH)$_{3(s)}$). Indeed, the transformed curves point to buffer zones at pH values corresponding to the dissolution of these minerals (Figure 3a). The integration of the transformed curves indicates the amount of protons required to counter the buffer zones, and scores a total value of 0.0921 mol/L. This addition corresponds to a respective dissolution of orthoclase, kaolinite, and iron hydroxides of 0.0908, 0.001222, and 0.0010 mol of minerals per liter of solution. These amounts are equal to the dissolution of 8% and 21% of the initial content, respectively, of orthoclase and kaolinite. The latter is potentially a secondary mineral resulting from the weathering of Al–silicate phases. The dissolution of these phases consumes protons, as explained by the following reactions (Equations (6)–(8)):

$$KAlSi_3O_{8(s)} + 4\,H^+ \Leftrightarrow Al^{3+} + K^+ + 2\,H_2O + 3\,SiO_2 \tag{6}$$

$$Al_2Si_2O_5(OH)_{4(s)} + 6\,H^+ \Leftrightarrow 2\,Al^{3+} + 2\,SiO_2 + 5\,H_2O \tag{7}$$

$$Fe(OH)_{3(s)} + 3\,H^+ \Leftrightarrow Fe^{3+} + 3\,H_2O \tag{8}$$

It is also important to note that iron hydroxides impact the pH through their dissolution and surface complexation sites, as illustrated below in Section 3.3.2 (see Equations (14) and (15)).

The main phases that impact the pH in the acidic zone of the Mibladen mining waste are dolomite (CaMg(CO$_3$)$_{2(s)}$), calcite (CaCO$_{3(s)}$), fluorite (CaF$_{2(s)}$), and iron hydroxides (Fe(OH)$_{3(s)}$). Dolomite and calcite buffer the solution at a pH of approximately 6 (at respective values of 5.8 and 6.5; see Z4 and Z5 in Figure 3b). Fluorite adds a buffer zone at a pH of 3.4 (see Z4 in Figure 3b). The integration of the transformed curves indicates that it

takes 0.114 mol of acid to pass the first buffer zone (Z5) and 0.024 mol of additional acid to pass the next buffer zone (Z4). This corresponds to the dissolution of the major phases of 0.017 mol of dolomite, 0.013 mol of calcite, and 0.000082 mol of fluorite per liter of solution, i.e., 8%, 7%, and 0.2% of their initial concentrations. The reaction of mineral dissolution by the consumption of protons can be expressed as follows (Equations (9)–(11)):

$$CaMg(CO_3)_{2(s)} + 2\,H^+ \Leftrightarrow Ca^{2+} + Mg^{2+} + 2\,HCO_3{}^- \tag{9}$$

$$Ca(CO_3)_{(s)} + H^+ \Leftrightarrow Ca^{2+} + HCO_3{}^- \tag{10}$$

$$CaF_{2(s)} + H_2O + H^+ \Leftrightarrow CaOH^+ + 2\,H_2F_2 \tag{11}$$

These findings show that the model appropriately captures the bulk buffer capacity. They also show how the geochemical model can be used (and was used) to derive the composition of the mineral assemblages. In more concrete terms, the transformed curve can be aligned with several models simulating the dissolution of plausible mineral phases to detect the phase that corresponds the best. Such a method may allow for the identification of each of the active mineral phases, buffer zone by buffer zone, leading to the final composition.

### 3.2.3. Validation of the Proposed Mineral Composition with the Tailings' Characterization

The final composition of the mineral assemblages deduced from inverse modeling (Table 2) agrees well with the characterization of the mine tailings (using XRD analysis and SEM). Indeed, most of the phases identified by the inverse modeling (Table 2) were detected either by XRD (Table 1) or SEM (see Section 3.1). In particular, pyrite, which is crucial regarding geochemical processes in mine tailings and identified by inverse modeling, was also recognized by SEM for the two mine tailings. This consistency between mineral characterization and inverse modeling consolidates the proposed final composition of the mineral assemblage. However, a few minerals had to be considered, even though they were not detected either by XRD or by SEM (i.e., alamosite, hematite, and fluorapatite). Numerical sensitivity analysis proved that these phases were needed to fit the observed evolution of pH with Eq_H$^+$ accurately. Conversely, other minerals were detected with SEM and are not part of the final mineral assemblage identified by inverse modeling. For instance, massicot (PbO$_{(s)}$) was observed with the MEB but is not part of the final assemblage (see Table 2).

Several reasons may explain the discrepancy between mineral characterization and ANC–BNC experiments. First, the differentiating step that allows for the detection of buffer zones may be impacted by the lack of precision of the experimental pH curves (Eq_H$^+$). Uncertainty in the measure of pH and the amounts of added protons may affect the quality of the derivative. Second, ANC–BNC experiments detect the main contributors to the bulk buffer capacity. Minerals with meager concentrations may contribute to a tiny proportion of the bulk buffer capacity, making them undetectable. Third, ANC–BNC experiments do not involve strong enough conditions to dissolve all the minerals, with most of them requiring very acidic conditions along with high temperature and pressure conditions to dissolve and thus exhibit any buffering capacity [44,45]. Despite potentially significant concentrations, these minerals cannot be detected using ANC–BNC experiments. Fourth, ANC–BNC experiments are conducted over reasonable durations (48 h), allowing only the quickest processes to occur. Kinetically limited dissolution processes do not have enough time to complete. Consequently, ANC–BNC must be combined with additional information to complete the characterization of the solid phase. Our study combined our data with the knowledge derived from previous work to obtain the most probable composition of the mining waste (see Table 2).

### 3.2.4. Comparison of Waste from the Two Mining Sites: Two Types of Mine Drainage

The geochemical model reveals the common and contrasting features between the two mine tailings. While only a tiny amount of alkaline solution is needed to reach the

asymptote at pH = 12, more acid is required to reach the final asymptote in the acid zone at pH = 2.35 in the waste from the two mining sites. For Zeida mine tailings, it takes less than 0.07 mol, while it takes more than 0.15 mol for Mibladen tailings (Figure 2). Such a difference indicates a significant difference in terms of mineral composition. The waste from both mining sites is composed of the same types of mineral phases (including sulfides), except that the Mibladen tailings are strongly carbonated. The presence of carbonates (calcite and dolomite) explains the significant difference observed in the ANC–BNC experiments, particularly the tremendous buffering capacity zone observed between pH 5 and 7 (Figure 3). Calcite and dolomite are expected to make a significant difference during weathering processes and related mining drainage. Their buffer capacities are expected to counteract the acidification produced by the weathering of sulfide phases such as pyrite ($FeS_2$). In both mine tailings, thanks to SEM analysis, we also detected traces of pyrite (see Table 2), the impact of which is so significant that even small quantities may significantly acidify the environment. However, calcite and dolomite in the Mibladen tailings' assemblage are expected to buffer such acidification processes and to lead to contaminated neutral drainage. Conversely, acid mining drainage is expected for the Zeida assemblage.

The investigation of waste from the Zeida and Mibladen mining sites showed the two main drainage scenarios: acid mine drainage (AMD) and contaminated neutral drainage (CND) [1,39]. AMD, the best-known phenomenon, is due to autocatalysis by ferrous iron of the dissolution of pyrite that triggers very strong acidification of the solution [46]. On the other hand, CND occurs under basic conditions when some carbonate phases are in sufficient quantity to buffer the solution while dissolving the sulfide phases [39]. The type of mine drainage is expected to dictate the pollutant release [1,47,48].

*3.3. Mobilization and Speciation of Lead*

3.3.1. Mobilization of Lead as a Function of pH

The samples from the studied sites contain lead in significant quantities. The Zeida mine samples are the least loaded, with total concentrations of lead between 3610 and 4970 mg/kg. The Mibladen mine samples are more concentrated, containing between 4640 and 5140 mg/kg of lead [1]. We remind the reader that the samples were homogenized for each mine tailing before performing the ANB-BNC experiments.

First, in the neutral zone, when the pH is between 6 and 8, lead is very poorly mobilized (Figure 4). Conversely, lead solubilization is greatest in the acid zone for the two samples (Figure 4). In addition, lead solubilization is significantly greater in Zeida samples in comparison to Mibladen samples. In terms of proportion, the mobilized quantities approach 77% on average for Zeida, compared to 1% for Mibladen of the total lead content. This difference in lead mobilization depends mainly on the carrier phases but also the pH. Due to a lower buffering capacity, lower pH values were attained for Zeida (Figure 2a vs. Figure 2b). Further acidification (by providing larger amounts of protons) may have increased the quantity of lead mobilized for Mibladen. However, we decided to apply the same standardized protocol to the two different types of mining waste. By doing so, we simulated experimentally similar conditions of exposure to air and water to be more representative of actual environmental conditions (leaching by acid rain or even the use of waste in the construction industry). By revealing this significant difference between the two types of mining waste, we could demonstrate the effect of dolomite and calcite on acidification and lead mobilization. However, the increased buffering capacity of Mibladen is not sufficient to completely prohibit lead mobilization.

As stated above, given the agreement between observed and modeled data (see Figure 4 and Section 3.1.2), the model is considered as representative of lead mobilization. Thus, it is possible to conclude on geochemical processes and lead speciation in both the liquid and the solid phases based on the modeled results, at least for the studied closed reactors. The extension of our findings to the prediction of lead release in the field is addressed in Section 3.4.

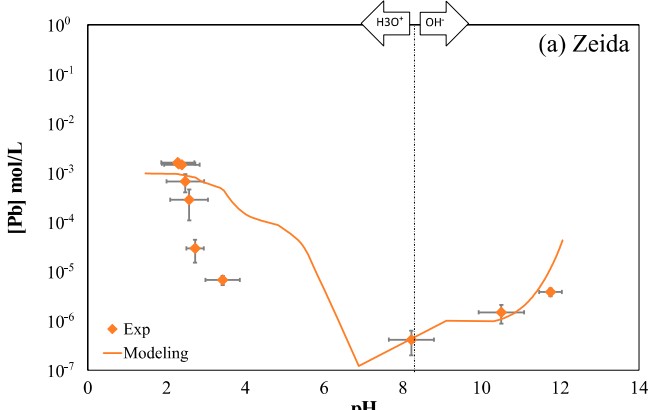
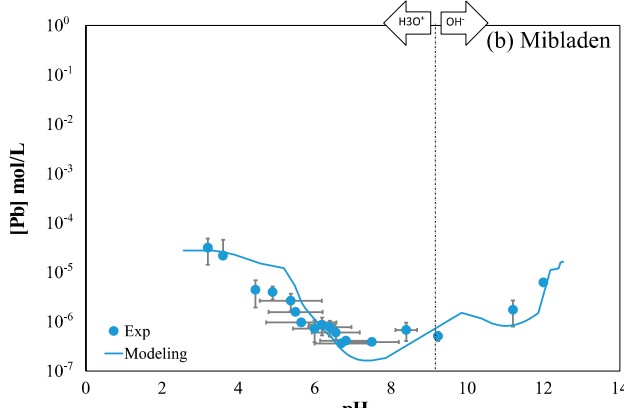

**Figure 4.** Solubilization of lead (total dissolved concentration), as a function of pH, from Zeida (**a**) and Mibladen (**b**) mine tailings; experimental curves (cross, adapted from [1,32]) and geochemical modeling (solid line).

### 3.3.2. Speciation of Lead as a Function of pH

The benefit of the modeling approach is fully illustrated by the provision of the characterization of lead speciation as a function of pH (Figure 5). The study here focuses on the lead carrier phases that are impacted by the ANC–BNC tests. The other carrier phases that are slightly or not impacted by the pH are not represented here since they play no role regarding lead mobilization. For the sake of precision, the whole range of pH meshes into 200 increments in the acidic zone, 200 increments in the basic zone, plus 1 point for the neutral zone (contact with distilled water). For all of these modeled points, we provide the mineral composition of the solid and liquid phases and lead speciation, including lead in solution, lead sorbed onto surface complexation sites (iron hydroxides) or cationic exchange sites, and lead precipitated or associated with lead-bearing minerals (Figure 5).

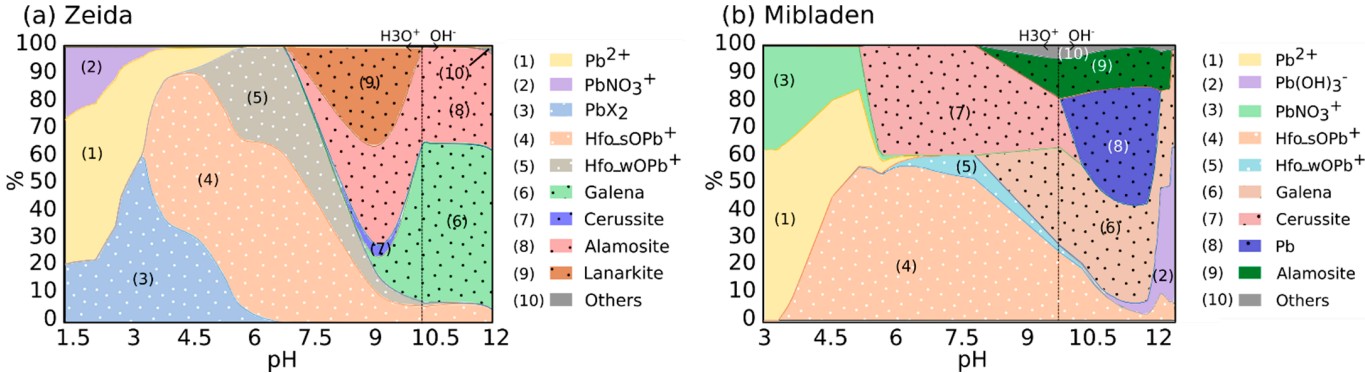

**Figure 5.** Speciation of lead, as a function of pH, for Zeida (**a**) and Mibladen tail (**b**) mining waste, obtained using the geochemical model (black dots: crystalline phases, white dots: surfaces, plain: in solution). Note that the range of the *x*-axis corresponds to the range of pH obtained for the same ANC–BNC tests with the same amounts of acidity and alkalinity. Consequently, the pH ranges differ between the two cases.

In Figure 5, the crystalline phases carrying lead are represented by black dots, while the iron hydroxide phases and ion exchange phases are represented by white dots. Lead in solution is represented by a solid color. The mineral phases that impact the mobilization of lead are galena, cerussite, alamosite, and lanarkite. These phases are relatively stable in neutral and acidic zones and control the mobilization of lead in these zones. In Mibladen tailings, calcite and dolomite promote the precipitation of calcareous phases such as cerussite, which plays a more important role here than it does in Zeida tailings (Figure 5b vs. Figure 5a).

Lead sorption to hydroxides and cationic exchange sites also contributes significantly to its speciation. This is particularly clear for Zeida (Figure 5a). Iron hydroxides sorb lead onto strong (s) and weak (w) sorption sites, referred to as $\equiv$Hfo_sOH and $\equiv$Hfo_wOH, respectively:

$$\equiv\text{Hfo\_sOH} + \text{Pb}^{2+} \Leftrightarrow \equiv\text{Hfo\_sOPb}^+ + \text{H}^+ \tag{12}$$

$$\equiv\text{Hfo\_wOH} + \text{Pb}^{2+} \Leftrightarrow \equiv\text{Hfo\_wOPb}^+ + \text{H}^+ \tag{13}$$

$$\equiv\text{Hfo\_sOH} \Leftrightarrow \equiv\text{Hfo\_sO}^- + \text{H}^+ \tag{14}$$

$$\equiv\text{Hfo\_wOH} \Leftrightarrow \equiv\text{Hfo\_wO}^- + \text{H}^+ \tag{15}$$

When present in the solid phase, iron hydroxides can sorb lead. Conversely, they release the sorbed lead when they dissolve. These results are in agreement with the literature, and sequential extractions were carried out on these types of matrices. Previous works attest that lead is mainly linked to the crystalline fraction and to iron hydroxides [49,50]. The distribution and the impact of mineral assemblages on mobilization depend very strongly on the total content, the type of soil, and its properties [49,51].

In solution, lead is not very well complexed. It is mainly found in the form of $\text{Pb}^{2+}$ in acidic conditions. Since nitrate is added when acidifying with $\text{HNO}_3$, lead binds to nitrate to form $\text{PbNO}_3^+$. In the basic zone, it is mainly associated with the $\text{OH}^-$ anion, in excess in solution, and forms $\text{Pb(OH)}_3^-$. These results of lead speciation as a function of pH are consistent with the literature [52]. Figure 5 may indicate lead speciation in the field, for any given pH, provided that ANC–BNC results are representative of element release in the field (this point is addressed in Section 3.4). This work's significant contribution is that it provides more insight on the lead speciation that can be combined with experimental methods of sequential extraction. These are known to be insufficient for a comprehensive understanding of all the mechanisms involved in lead and trace metal mobilization [53].

These results are representative and consistent with the phenomena of mine drainage, both for the levels of mobilization of lead and speciation in the liquid and solid phases as a function of the pH. We have significant amounts of lead in both Zeida and Mibladen mine tailings, meaning significant pollution (Figure 4). These two examples represent two distinct cases of mine drainage, i.e., AMD and CND. AMD is the most well-known phenomenon that impacts the mobilization of elements and produces very high acidity. This acidity logically promotes the release of trace metals. However, CND also impacts the chemical quality of the leachate even though the buffering capacity minimizes acidity. Indeed, even if calcareous phases buffer the solution, they do not prevent the dissolution of sulfides and other lead-carrying phases, resulting in lead release [1,47,48]. In these two conditions (AMD and CND), iron hydroxides appear to be a crucial contributor to lead release.

### 3.4. Representativeness of ANC–BNC Experiments

ANC–BNC runs were performed to characterize pH evolution and element release (lead) as a function of the amounts of added acidity or alkalinity, as suggested by [39] for mine tailings. Geochemical modeling was deployed to determine the amounts of minerals that were involved in the observed pH trends and element release. The mineral assemblages were characterized in terms of selected minerals and related amounts and validated against mineral characterization (XRD analysis and SEM). The number of sorption sites onto iron oxides and clay (cationic exchange) was also deduced from fitting experimental data, whereas the equilibrium constants were fixed at their regular values (the *llnl* database). From these data, speciation panels were proposed as a function of pH (Figure 5), assuming that the knowledge of pH was enough to derive the fate of lead in those mine tailings under all circumstances.

It should be noted that these modeled data were inferred from modeling specific ANC–BNC experiments that quantify buffer capacity and element release under specific

experimental conditions. These may significantly differ from field conditions with a collection of contrasting conditions [54], including the following features:

- In the field, the contact with the atmosphere strongly differs between field and batch conditions, with closed conditions for batch experiments versus utterly open conditions (contact with a quasi-infinite reservoir of gas). In the case of batch reactors, only the gaseous species present in the volume of gas may interact with solutes. The consumption of oxidants (such as oxygen) may limit oxidation processes, whereas oxidants are instantaneously renewed in the case of contact with the open atmosphere. We then expect more pyrite oxidation [55] and induced acidification, mineral dissolution, and lead release under field conditions (no limitation of $O_{2(g)}$), e.g., [2,53]. In other words, if the three phases remain the same between batch and field conditions, the gas phase does not have the same composition.
- In the field, very slow processes that are kinetically limited may govern element release for a very long period of time, whereas the contact time in batch reactors is restricted to a few days, thus selecting mostly the instantaneous mechanisms. The addition of acid or alkalinity that is performed to boost dissolution and element release may not compensate for the very long time needed for some minerals to dissolve. For instance, we expect pyrite dissolution to be kinetically limited [56,57] and thus much less important under batch conditions than under field conditions. The same expectations may be stated for calcite and dolomite dissolutions that are prone to significant kinetic limitations [58].
- The liquid/solid ratio is all but the same between batch and field conditions. Indeed, in batch reactors, the solid phase is dispersed into the liquid solution with a very high liquid/solid (L/S) ratio, in the order of 10 L/kg in our cases and in most studies [54]. In the field, assuming usual values of soil bulk and mineral densities, i.e., 1.75 g/cm$^3$ and 2.65 g/cm$^3$, nominal values of saturated water content and the corresponding L/S ratio can be estimated, revealing a value of 0.2 L/kg in the field versus 10 L/kg in the batch reactors. We expect that higher L/S ratios would promote solubilization and element release.
- In addition to the difference in the liquid/solid ratio, field conditions are characterized by dynamic conditions (flow-induced solute transport), whereas batch reactors simulate static conditions with optimal contact between the solid grains and solutes. Numerous previous works have already demonstrated that the elements' fate significantly differs between static and dynamic conditions [59–61]. In columns, the local hydrodynamic conditions make the access of solutes to sorption sites more difficult and thus may derive from optimal sorption [62]. In addition, the soil structure may be prone to preferential flows, in particular for soils with aggregates and macropores, e.g., [63,64], which may, in turn, reduce the access of solutes to the reactive particles and decrease solute sorption [65–67].

In conclusion, ANC–BNC experiments combined with modeling can be seen as a promising tool for characterizing mineral compositions and sorption sites. However, it should not be considered the only representative solution of element release in the field. It provides generic trends, and proved in our case that Mibladen mine tailings have a high carbonate content, which should counteract any source of acidification (including that provided by pyrite oxidation) and lower, without preventing it completely, the release of lead (and other trace metals). The prediction of metal trace release in the field should not be purely inferred from ANC–BNC experiments, but rather predicted by modeling the field's actual conditions and considering knowledge support using ANC–BNC experiments on mineral composition and sorption sites. For our specific study case, we may also advise conducting additional experiments or monitoring in the field to consolidate our findings.

## 4. Conclusions

This study's objective was to characterize the mechanisms of lead mobilization and its speciation as a function of pH in waste from two mining sites, representative of AMD

and CND, to better manage mining waste [68]. The main challenge was to set up and apply a methodology combining experimentation and modeling. This new and promising methodology provides crucial information for understanding the dominant mechanisms responsible for major and trace elements, including trace metal (e.g., lead) release and speciation in water. Such an approach has already been validated with pure phases mimicking mineral assemblages prone to generating AMD and CND, but it required further validation with real matrices [39].

The acid–base neutralization capacity tests revealed buffering powers specific to the two types of mining waste and strongly depended on carbonates. The geochemical modeling associated with the experimental results proved very satisfactory and confirmed our hypotheses based on our interpretation of the experimental results. The association between experimental results and geochemical modeling proved very relevant for studying the impact of solid phases on the buffering capacity, mobilization, and speciation of lead as a function of pH.

The carbonate phases (dolomite and calcite), which have a strong buffering capacity, are buffer acidity sources that do not totally prevent contaminated neutral drainage. The dissolution of the crystallized mineral phases containing lead and the desorption of lead from surface complexation sites (iron hydroxides) are the predominant mechanisms for lead release in solution. Despite different buffering powers, the mobilization of lead follows the same trends as a function of pH, but with much higher concentrations in the case of AMD. The results of lead speciation as a function of pH are consistent with the literature. Most notably, they allow for an accurate description of the evolution of the various lead-bearing phases as a function of pH. They also make it possible to determine the lead's speciation in the liquid phase and thus shed light on the potential toxicity of the leachate produced by mining waste.

This study provides a detailed understanding of the state of the system, assuming that ANC–BNC experiments simulate real conditions accurately. To better understand these mechanisms under natural conditions, it would be attractive to combine our findings with investigations adapted to dynamic conditions accounting for the kinetics of mineral dissolution and slow processes. Indeed, in the field, the contact between mine tailings, water, and atmospheric gases is much more complex with dynamic conditions, variable solid/liquid ratios, much longer contact times, etc. We may advise combining laboratory results with field monitoring to strengthen the statements on lead release and the related toxicity. In addition, while this study focuses on the mobilization of lead, other trace metals should be considered [33]. ANC–BNC and geochemical modeling appear to be very promising tools for the characterization of mining waste and related mining drainage. However, as with any tool, other types of information, in particular field investigations, and other information sources should be considered as well.

**Author Contributions:** Conceptualization, C.D. (Clémentine Drapeau), L.L., and C.D. (Cécile Delolme); methodology, C.D. (Clémentine Drapeau), L.L., R.A., M.B., R.H., and C.D. (Cécile Delolme); software, C.D. (Clémentine Drapeau) with the help of L.L., C.D. (Clémentine Drapeau), D.B., M.E., M.B., and T.B.; validation, C.D. (Clémentine Drapeau), L.L., and C.D. (Cécile Delolme); resources, C.D. (Cécile Delolme) and L.L.; data curation, R.A., R.H., M.B., and C.D. (Clémentine Drapeau); writing—original draft preparation, C.D. (Clémentine Drapeau) with the help of L.L. and C.D. (Cécile Delolme); writing—review and editing, all authors. All authors have read and agreed to the published version of the manuscript.

**Funding:** This work received the support of the INFILTRON project funded by the French National Research Agency (ANR-17-CE04-010).

**Institutional Review Board Statement:** Not applicable.

**Informed Consent Statement:** Not applicable.

**Data Availability Statement:** Not applicable.

**Acknowledgments:** The authors would like to thank the reviewers and editors for improving the manuscript.

**Conflicts of Interest:** The authors declare no conflict of interest.

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
