# Peer review of "Lead Mobilization and Speciation in Mining Waste: Experiments and Modeling"

_minerals, doi:10.3390/min11060606_

Round 1

Reviewer 1 Report

Minerals 1159957

Lead mobilization and speciation in mining waste: experiments and modelling

This is a good piece of work and can be published after thorough language editing and minor corrections stated below.  

  1. The article is lacking good introduction that is relevant to the main composition of AMD, tailing and the reason why the study emphasis on lead mobilisation in the introduction part. In fact, he information has been given under section 3.2.1, particularly the concentration of lead given need to be taken to the introduction part and clearly stated in one paragraph at the end of the introduction.
  2. Table 1: Note that the XRD % composition of minerals is relative to crystalline minerals available in the sample but not relative to the total mass because some of them are in an amorphous form and other in the elemental form not detected. Such information needs to be included in the manuscript.
  3. 3 Y-axis write as: Pb2+Con.  (mol/L)

Author Response

Dear reviewer,

many thanks for your suggestions.

Please, find enclose our answers.

Best regards

Laurent Lassabatere

Reviewer 2 Report

Review of “Lead mobilization and speciation in mining waste: experiments and modeling” by Clémentine Drapeau , Rabei Argane, Cécile Delolme, Denise Blanc, Mostafa Benzaazoua, Rachid Hakkou, Thomas Baumgartl, Mansour Edraki and Laurent Lassabatère.

General Comments:

This paper presents geochemical analysis of tailings samples collected at two mine sites in Morocco. Analysis performed includes XRD, SEM-EDS for characterization of mineralogy and an ANC-BNC test (essentially acid titration).  The mine wastes show different characteristics with one waste rich in carbonates (Mibladen) and the either literally devoid of carbonates (Zeida). The study focuses on the response of these materials to the tests and the mobility of the trace element Pb. The geochemical analysis is accompanied by geochemical modeling using PHREEQC, which in this context is used to simulate the ANC-BNC tests, and aid in the interpretation of the results. The experimental and modeling techniques are fairly well described and sufficient background and context for the study is given. The materials react as one would expect, with the carbonate-rich mine waste providing strong pH-buffering capacity and attention potential for Pb, while the material without any significant carbonate content shows little pH-buffering capacity and a high potential for Pb mobilization. This result is not new. It is common knowledge and an expected outcome. The modeling results provide a good fit to the experimental data for pH and Pb and based on this outcome the authors drive pH-buffering mechanisms and Pb speciation for the samples as a function of pH. Although this work is interesting and the fits are good, it is not clear to me whether these results are representative of conditions in the tailings. The ANC-BNC test is a short term test and did not evaluate the effect of sulfide oxidation. Instead, acidity was added. This approach provides information on pH-buffering capacity, but not on the acidification that would occur under natural conditions, which is more a function of sulfide content. In addition, I have several questions about the interpretation and findings, as outlined below. Although the fit is good, it is limited to pH and Pb and aspects of non-uniqueness have not been addressed. I am uncertain, if the results re pH-buffering by silicate phases for Zeida material will hold and whether the predicted Pb speciation as a function of pH would actually occur as predicted. There is no experimental evidence supporting the Pb speciation. I don’t think that the geochemical modeling allowed for the “accurate identification” of mineral phases.   

Specific Comments:

Lines 112-113: This is a fairly small data set. Is it presentative for the tailings impoundment? What was the size (volume) of each sample? How was the sample mixed? How were subsamples extracted for mineralogical analysis?

Lines 125-126: Did cerussite occur in both materials? Is it a primary phase? It could be a secondary phase in Mibladen tailings, since the sample was extracted from weathered tailings.

Line 135: metallic traces? – do you mean trace metals?

Line 149: acid solutions of 1.5 mol/l and 0.5 mol/l. Of what?

Line 164: de l’USGS – change to English for consistency with rest of article

Line 192: 700 and 900 simulations is a lot. Was this done manually? Can anything more specific be said about arriving at the final solution. Why was this particular solution chosen?

Lines 203-204: Were minerals such as alamosite, lanarkite and massicot identified in the SEM-EDS analysis. The tailings have already weathered for a while at these abandoned mines. If these phases do play a significant role, one would expect that they can be identified via SEM.

Table 2: The initial mineral contents seem very low in comparison to what pore water would see. I think this is due to the nature of the ANC/BNC test. What are the implications on the representativeness of the results?

Table 2: “Gas phases” , not “Gaz phases”

Table 2: “Mineral phases bearing lead”

Table 2: What is done here with oxygen. Why is oxygen expressed in terms of N species??? Why is the partial pressure of O2 set to 1 atm.

Table 2: The parameters Vol_Air_fix is not explained, even in the footnote, what does it do?

Table 2 footnote: “fix”, not “fox”

Table 2: where does the surface site data come from for the surface complexation and ion exchange reactions? No data is given. No source. Also not determined with the inverse model.

Section 3.1.1 “initial thought” not appropriate for a journal paper

Lines 217/218: I agree that the fit is good, but what about model non-uniqueness. The model contains calibrated and assumed parameters (see Table 2). There could be other phases. There could be different surface site abundance.

Lines 226ff: The model may be representative of processes occurring in the ANC/BNC tests, but not necessarily in the tailings. There is no pH data from the tailings, just the mineralogical data from the 7 samples. Processes in tailings will be driven by the oxidation of sulfides. Little is said about sulfides in this paper and their oxidation is not assessed in the experiments and in the modeling.

Lines 236-237. I strongly doubt that CO2(g) will lead to acidification. There are bigger players at work (oxidation of sulfides). In many cases, CO2 is released from tailings in response to pH-buffering processes. This has the opposite effect (removal of acidity via CO2 degassing. I guess the fixed gas volume will come into play here. The role of this formulation is unclear.

Line 267: I strongly doubt that quartz provides a pH buffer effect over a time frame of only 48 hours. Its kinetics are very slow, even under strongly acidic conditions. It is likely that other processes are at work.

Line 271: “buzzer zones”?

Line 274: kaolinite is a buffer, usually a secondary phase in these environments in response to Al-silicate weathering.

Lines 276-289: only a relatively small fraction of the primary phases become depleted in the experiments. Why does the pH-buffering stop then and move on to the next phase. Normally, the buffering phase becomes depleted. Why does pH become quite low for the Mibladen samples? I consistent of more than 50% carbonates?

Line 318: Mibladen tailings in brackets not needed.

Lines 320-321: This is well-known information. I would not consider this a main finding worth discussing at length.

Line 324: and/or

Lines 340 ff: General results for Pb for these two materials are as expected, not a surprising or new finding.

Lines 358-360: A well-known fact. Pb mobility in carbonate rich material is very limited due to precipitation of hydroxides, carbonates and adsorption.

Lines 365-367: How is the titration model representative for field conditions? It addresses the pH-buffering capacity, but not the release capacity of the material due to sulfide oxidation (galena, pyrite, etc…). Maybe these processes have already gone to completion in these old tailings?

Figure 4: It is unclear how the predicted Pb speciation is informative for Pb release from tailings. No information is provided on aqueous tailings geochemistry.

Line 391: Why would galena control Pb mobility. It is a source for Pb, since it is undergoing oxidation.

Section 3.3, Table 3: Goodness of fit is a necessary criterion for model validity, but it is not sufficient. Model non-uniqueness play also a big role. Overall, the model was only constrained by pH and Pb. Other geochemical data was not shown. It is quite possible that the fit for pH and Pb is good, but it’s not good for other anions and cations. What does that mean for the model?

Author Response

Dear reviewer,

many thanks for your suggestions. We appreciated your extensive review.

Please, find enclosed our answers.

Best regards

Laurent Lassabatere

Reviewer 3 Report

Dear authors:

In my view, the manuscript can be accepted in the present form.

Authors provide substantial information and further analysis to a complete understanding of material. As a final sugestion, the authors are invited to verify stoichiometry of compounds mentioned.

Author Response

Dear reviewer,

many thanks for your comments and your time.

We searched the manuscript for typos. Thanks again.

Best regards,

Laurent Lassabatere on behalf of the authors

Round 2

Reviewer 2 Report

The paper has improved and can now be accepted for publication.

Author Response

Dear reviewer,

many thanks for your comments and your time.

Best regards,

Laurent Lassabatere on behalf of the authors